# Invited perspectives: challenges and future directions in improving bridge flood resilience

[1]Enrico Tubaldi, [1]Christopher J. White, [1]Edoardo Patelli, [2]Stergios Aristoteles Mitoulis, [3]Gustavo de Almeida, [4]Jim Brown, [5]Michael Cranston, [6]Martin Hardman, [7]Eftychia Koursari, [8]Rob Lamb, [5]Hazel McDonald, [9]Richard Mathews, [10]Richard Newell, [11]Alonso Pizarro, [12]Marta Roca, [13]Daniele Zonta

[1] University of Strathclyde, Glasgow, UK
[2] University of Surrey, Guildford, UK
[3] University of Southampton, Southampton, UK
[4] Transport Scotland, Glasgow, UK
[5] Scottish Environment Protection Agency, Perth, UK
[6] Cumbria County Council, Carlisle, UK
[7] Amey, Glasgow, UK
[8] JBA Trust, UK and Lancaster University, Lancaster, UK
[9] Mott MacDonald, Altrincham, UK
[10] Network Rail, Milton Keynes, UK
[11] Universidad Diego Portales, Santiago, Chile
[12] HR Wallingford, Wallingford, UK
[13] University of Trento, Trento, Italy

*Correspondence to*: enrico.tubaldi@strath.ac.uk

**Abstract.** Bridges are critical infrastructure components of road and rail transport networks. A large number of these critical assets cross or are adjacent to waterways and floodplains and are therefore exposed to flood actions such as scour, hydrodynamic loading and inundation, all of which are exacerbated by debris accumulations. These stressors are widely recognised as responsible for the vast majority of bridge failures around the world, and they are expected to be exacerbated due to climate change. While efforts have been made to increase the robustness of bridges to the flood hazard, many scientific and technical gaps remain. These gaps were explored during an expert workshop that took place in April 2021 with the participation of academics, consultants and decision makers operating mainly in the United Kingdom and specialising in the fields of bridge risk assessment and management and flood resilience. The objective of the workshop was to identify and prioritise the most urgent and significant impediments to bridge flood resilience. In particular, the following issues, established at different levels and scales of bridge flood resilience, were identified and analysed in depth: (i) characterization of the effects of floods on different bridge typologies, (ii) uncertainties in formulae for scour depth assessment, (iii) evaluation of consequences of damage, (iv) recovery process after flood damage, (v) decision-making under uncertainty for flood-critical bridges, and (vi) use of event forecasting and monitoring data for increasing the reliability of bridge flood risk estimations. These issues are discussed in this paper to inform other researchers and stakeholders worldwide, guide the directions of future research in the field, and influence policies for risk mitigation and rapid response to flood warnings, ultimately increasing bridge resilience.

**Keywords:** Flood risk, bridges, resilience, decision-making, scour, vulnerability, monitoring, forecasting.

## 1. Introduction

Bridges are critical infrastructure components of road and rail transport networks. A large number of these critical assets cross or are adjacent to waterways and floodplains and are therefore exposed to river flooding actions such as scour, inundation and debris impact. The hydraulic risk of bridges to flood impacts is significant globally. The United Kingdom is a country where floods are of particular concern for bridge safety and operability due to the high frequency of extreme hydrometeorological events, and the significant cascading impacts of the failure of these critical assets on wider transport networks, communities and businesses. The United Kingdom does not have a national structures database. The number of bridges managed by the Highways Agency is estimated to be as high as 160,000, with approximately 30,000 of these crossing waterways (Middleton 2004). Network Rail also manages over 8,800 bridges in or adjacent to inland waterways (Lamb et al., 2019). While these estimates are uncertain and only encompass the main bridge asset managers in the country, they give an idea of the high exposure of bridges to the flood hazard. The 2009 flood event in Cumbria alone resulted in 29 road bridge collapses or severe damage, £34m in repair and replacement costs, and significantly larger economic and societal costs (Argyroudis et al., 2019). The December 2015 floods have also resulted in further losses, amounting to approximately £25m direct costs. Railway bridges are also severely affected by floods, with 138 failures of these assets caused by flood-induced scour in Britain between 1846 and 2013 (Van Leeuwen and Lamb, 2014). Passenger travel disruptions due to floods were estimated to cost up to £60 m/yr for the UK railway network alone (Lamb et al., 2019), and indirect losses (e.g. impact on economic productivity) can be over one order of magnitude larger. These numbers provide a measure of the costs incurred by councils, transport operators and businesses due to floods.

The fact that bridges continue to fail at a very high rate and the severe disruptions caused by bridge closures due to floods demonstrates the issues and uncertainties associated with current procedures and practices for assessing and mitigating the flood risk. These issues are due to a combination of factors, among which the lack of knowledge of the problem, the gaps existing between the advanced techniques and methodologies developed by researchers and the more practical approaches adopted in risk management procedures, the lack of adequate human and technical resources, significant budget constraints, the tendency to acknowledge and address issues only when they manifest themselves in a catastrophic manner and to suppress rather than resolve problems. An analysis carried out by the RAC Foundation (2021) on bridges managed by local highways authorities in Great Britain has shown that there has been an apparent large decline in the number of bridges being assessed for risk of damage caused by river flow, despite 10 bridges fully collapsed and 30 partially collapsed in 2020. Thus, it is not surprising that the level of risk of many bridges exposed to flood effects remains largely unknown, with risk ratings still missing for many structures on secondary routes (more than 1000 structures in Cumbria County alone).

While efforts have been made to increase the robustness of bridges to withstand flood actions, transportation infrastructure managers face a unique challenge to prevent additional economic damage, often using maintenance budgets that are already stretched. For example, Transport Scotland spends £3-5m per annum on flood repairs and resilience works. The estimated cost to retrofit the 3,105 bridges managed by local councils classified as "substandard" is approximately £1 Billion (£985 million). However, budget restrictions mean that only 392 of

these substandard bridges will likely have the necessary work carried out on them within the next five years (RAC Foundation 2021).

The projected increase in winter precipitation and river flows due to climate change is expected to increase further the risk of bridge failure due to flooding (Jaroszweski et al., 2021). This issue is also exacerbated by the long service life of bridges, often exceeding the design values of 50-100 years, implying that many bridges were built long time ago, with no consideration of the impact of climate change on the intensity of flood actions.

Responding to the challenges posed by the river flood hazard to bridges requires quantified cost/benefit analyses of both capital maintenance/mitigation and emergency response strategies. Moreover, a joint effort of academics with different backgrounds (e.g. hydraulic and structural engineers, hydrologists, etc.), decision makers from environmental and transport agencies, consultants from regional and local authorities, and technical specialists is required. This joint effort is necessary to fully exploit the advances in the various disciplines that thus far have worked in isolation.

In April 2021, an online workshop led by the University of Strathclyde was organised in conjunction with the University of Surrey and the University of Southampton. The workshop brought together experts from academia, consultants from engineering firms, managers from transport and environmental agencies and from councils operating mainly in the United Kingdom. The participants discussed and exchanged opinions, practices, experiences to identify research gaps and needs for the management and mitigation of the risk of bridge failure due to floods. The workshop was also organised to disseminate the latest research developments by the academics and to discuss and prioritise the existing needs and requirements by the industry and agencies.

The workshop structure and activities were carefully planned to maximise engagement and exchange of information. Supporting material was distributed before the workshop to stimulate thoughts and ideas and prepare the experts to actively participate in the discussion instead of trying to digest the information. During the workshop, each participant reported back their opinion and challenges with follow-up questions and plenary discussion. It is noteworthy that not all the partners could attend the workshop, and for this reason, some follow-up meetings were organised with some of the co-authors of this study. The meetings and workshops were complemented by further exchanges of emails and through feedback on an online document, where the partners shared additional thoughts and insights. At the end of this process, the exchanged ideas and expert opinions were aggregated, discussed, and summarized in the paper. The ultimate goal of this Invited Perspectives article is to inform other researchers and industry stakeholders worldwide, guide the directions of future research in the field, and influence policies for risk mitigation and rapid response to flood warnings, ultimately increasing bridge resilience in the United Kingdom and the rest of the world. Section 2 illustrates known challenges and knowledge gaps in both science and current risk management procedures. Section 3 provides general recommendations for future research in the field and for improving current emergency and risk management procedures.

## 2. Challenges and knowledge gaps

### 2.1. Flood actions on bridges and hydraulic modelling

Although scour is the most critical hydraulic action for bridges, other actions need to be considered in evaluating bridge flood risk, the most important being buoyancy (i.e. uplift forces exerted on submerged bridge components), hydrodynamic (drag) forces, impact forces exerted by large floating objects (e.g. vehicles). All of these

mechanisms are exacerbated by debris (e.g. wood) that accumulate around piers and decks during floods (Kirby et al., 2015, Mondoro and Frangopol, 2018; Cantero-Chinchilla and de Almeida, 2021). CIRIA Manual and the latest update (Kirby et al. 2015; Kitchen et al., 2021) provide an exhaustive state of knowledge on the assessment of debris impact and hydrodynamic forces on bridges, drawing on standards, guidance and research from various countries. It is worth to note that existing design guidelines for the assessment of hydrodynamic forces are non-conservative (i.e. do not provide an appropriate margin of safety) in some regimes, as demonstrated through an extensive experimental and numerical campaign by Oudenbroek et al. (2018a,b). These results have shown that important underestimation of forces may be obtained for cases in which free surface effects are important, for deeply submerged bridge decks, or for high blockage ratios. This and other recent studies have also highlighted that hydrodynamic forces can be significantly exacerbated by debris causing damming and build-up of water (also known as afflux). On a more practical level, transport agencies and operators stress the need for developing and/or reviewing the effectiveness of technical solutions for tackling the problem of mitigation of hydrodynamic forces for bridges at risk. Possible solutions could be aimed at reducing the hazard, by enlarging the cross-section area of the bridge, or by building a flood relief channel. However, both of these solutions can be expensive, disruptive and may have undesired morphodynamic consequences. Alternative solutions could be aimed at reducing the bridge vulnerability, by holding down the bridge deck onto the piers and the foundations in order to counteract the uplift action of water.

The characterization of the joint effects of the flood actions is complicated, and laboratory and numerical studies often focus on one or few specific actions (Ebrahimi et al., 2017). Experimental tests of the actual process of bridge failure have to address the challenge of scaling (Oudenbroek, 2018a,b), while computational analysis must overcome the issues related to modelling sediment transport and scour under complex, real-world conditions. Field measurements of all these simultaneous actions during floods are lacking, and studies deploying multiple sensors for monitoring both the bridge structure and the river flow are scarce (e.g. Crotti and Cigada, 2019). Moreover, typical models used for evaluating the hydraulic actions of interest often introduce some simplifications in the analyses, the limitations and impact of which on the results are not fully appreciated by end users. One example is the use of one-dimensional hydraulic modelling, which may not be suitable in the vast majority of real-world cases, where substantial gradients in the flow velocities are observed. In these cases, more accurate estimates of flood actions should be obtained resorting to two (or three)-dimensional modelling (e.g. Lai and Greimann, 2010).

## 2.2. Formulae for scour depth assessment

This subsection summarises the most critical issues and knowledge gaps in evaluation of scour at bridges that have emerged during the workshop. For a more detailed and exhaustive review of the problem, reference can be made to the CIRIA manual and subsequent updates (Kirby et al., 2015; Kitchen et al., 2021), and to the recent work of Pizarro et al. (2020a).

Typically, formulae for scour evaluation are based on laboratory flume tests at small scale to produce empirical relations between scour depth and parameters that can be controlled and measured in a flume, rather than seeking to establish the effect of parameters on the flow-field and the resistance of the bed sediment to erosion. While a wide range of conditions can be tested in flume experiments, very often tests adopt several simplifications of the

mechanisms operating at full-scale in natural rivers, e.g. steady or quasi-steady hydraulic conditions, uniform sediment sizes, and simple bridge and channel geometries. Whilst the flow-field around bluff surface piercing obstacles is complex, there is a need for a better understanding of the physics of local scouring around structures, and for developing general predictive models more strongly rooted on physical, rather than empirical grounds, as pointed out in Manes and Brocchini (2015).

Many empirical scour formulae are available that provide estimates of the equilibrium scour depth, which can be largely defined as the maximum scour depth that could be attained under a steady flow regime impinging the pier for a duration tending to infinity. Well-known equilibrium scour formulae are the Hydraulic Engineering Circular No. 18 (HEC-18, Richardson and Davis, 2001) and the Florida Department of Transportation (FDOT, Sheppard et al., 2014), which are widely used in the U.S.A. The scour manual by CIRIA (Kirby et al. 2015) suggests the use of the equation developed by Breusers (1977), which has later been further investigated (Melville and Sutherland, 1988; Breusers and Raudkivi, 1991; Melville and Coleman, 2000). Artificial intelligence (in particular Machine Learning) is increasingly being used to produce more accurate multi-variate empirical predictors for scour (see e.g. Sharafi et al. 2016). There is also a significant number of studies within the scientific literature comparing the accuracy of the formulae based on laboratory data and field data (see, e.g., Johnson et al., 2015; Liang et al., 2019; Park et al., 2017; Sheppard et al., 2014; Johnson et al., 2015; Qi et al., 2018; Wang et al., 2017; Shahriar et al., 2021). In general, application of common equilibrium scour formula results in significant overestimations of scour depths compared to field observations. This can be due to a number of reasons, including on site sediment non-uniformity, equilibrium scour depth not being attained, scaling effects inherent to flume experiments, complex pier and channel geometries compared to an idealized laboratory test, but also due to measurement inaccuracies. In addition, scour measurements are typically conducted after the flood event recedes, when the scour hole might have been refilled with sediment under live bed conditions (thus masking the maximum depth reached during the peak flow). Another significant source of uncertainty affecting the estimation of the maximum scour depth is the evaluation of the flow critical velocity separating clear-water from live-bed conditions (Hamidifar et al. 2021).

Methods for time-dependent scour evaluation have been developed that can be applied for the assessment of scour under single (or multiple) flood events, opening the avenues for more accurate scour estimates. The first studies on the topic considered the case of idealised hydrographs and clear-water conditions (see e.g. Oliveto and Hager 2002; Oliveto and Hager 2005), whereas more recent ones have also used more realistic hydrograph shapes. Recently, Pizarro et al. (2017a,b) and Link et al. (2017) proposed a model based on the dimensionless effective flow work, W*, for dealing with flood waves, and validated it against a wide range of unsteady conditions. Additionally, Link et al. (2020) proposed an extension of the model to consider the counter effects of erosion and deposition within the scour hole, which are typical of live-bed conditions.

The effects of debris on scour evolution are also a topic of extreme interest that has been subject of significant research efforts for many decades, since the early qualitative studies of Laursen and Toch (1956). Cantero-Chinchilla et al. (2021) lists the most important studies on the topic and presents an assessment of the influence of flow intensity, blockage area ratio, depth ratio on the development of local scour with flow-dependent debris accumulation. These parameters were found to be the most important ones also in other studies on the topic (e.g. Pagliara and Carnacina 2010), whereas the debris permeability, which affects significantly hydrodynamic forces,

has a minor influence on local scour (see also Lagasse et al. 2010). Debris accumulations can increase local scour depths by a factor of two or more compared to local scour depth without accumulations. The increase in scour depth that results from debris accumulations depends critically on the characteristics of debris accumulations (e.g. size and shape, which mainly determine their influence on scour) that will form at a given location, which is difficult to predict. Experiments by Panici and de Almeida (2018, 2020, Figure 1) provide methods to estimate

the maximum dimensions possibly formed under given flow and debris conditions. However, additional experimental research is needed to extend the range of applicability of existing methods and approaches, and to characterize the likelihood of accumulation of debris at bridge piers. Another topic that is receiving considerable attention by researchers is the pressure-flow scour due to vertical contraction, which takes place in the case of submerged bridge deck (Carnacina et al., 2019). A recent review paper (Majid and Tripathi, 2021) discusses the

many research needs in this field.

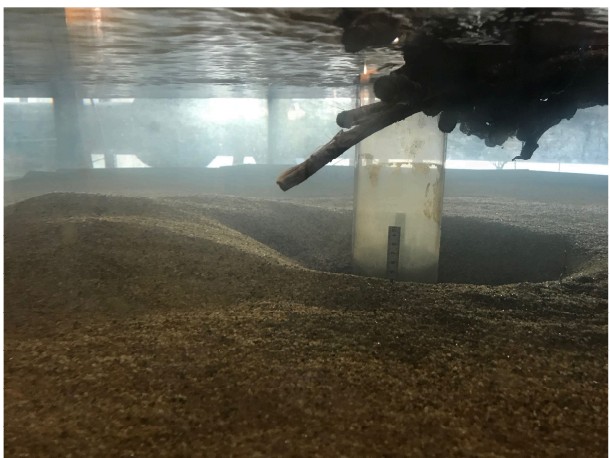

**Figure 1: Debris accumulation formed in the laboratory and developed scour hole (Cantero-Chinchilla et al. 2021).**

Another important issue that requires further investigation regards the prediction of the geometry of scour holes (and how it develops over time) for complex bridge pier geometries. It is usually assumed that the shape of scour hole is indeed independent of the flow conditions and that it can be approximated by an inverted paraboloid with the upstream slope corresponding to the sediment's angle of repose, but these assumptions work well only for simple geometries such as cylindrical piers, as proven by Chreties et al. (2013), local scour conditions, and also

for a flow direction perpendicular to the bridge longitudinal axis. Lee et al. (2021) have recently investigated experimentally the evolution of scour around piers and foundations with complex shape other than the cylindrical one, confirming that the maximum scour depth is attained upstream of the pier. The load bearing capacity of foundations and more in general the bridge response to scour and collapse mechanisms are significantly affected by the scour hole geometry (Maddison 2012). The numerical studies of Tubaldi et al. (2018) and Scozzese et al.

(2019) have shown how important it is to consider this when predicting or simulating the collapse behaviour of masonry arch bridges, exhibiting major damage in correspondence of their upstream side, where the scour hole is usually deeper (see Figure 2 and Figure 3). Thus, more research in this field is required in order to have an insight into the shape of scour holes that could develop at bridge foundations and under different flow conditions (e.g. angle of attack of flow).

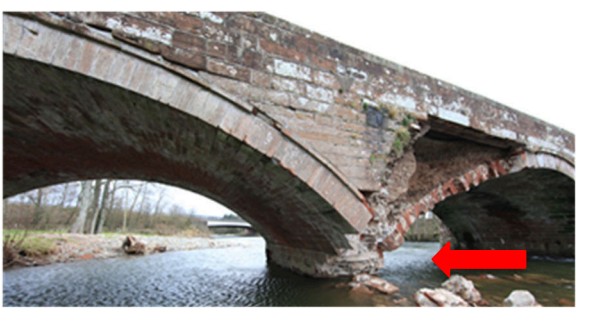 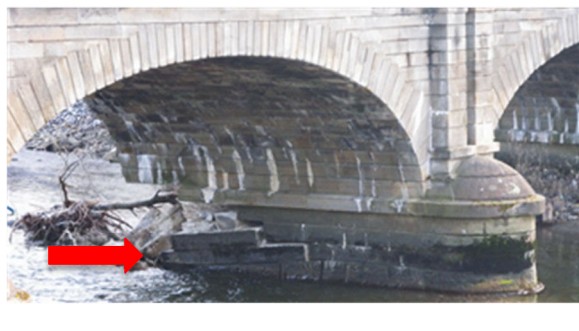

|(a)|(b)|
|---|---|

**Figure 2. Damage of Brougham Old Bridge (a) and of Calva bridge (b), typical of many masonry arch bridges subjected to scour (Source: Cumbria Council County for (a) and Bill Harvey for (b)). Flow direction indicated by arrow.**

### 2.3. Vulnerability of various bridge typologies

The evaluation of the vulnerability of bridges to floods has received little attention compared to other hazards such as earthquakes. This is mainly due to a combination of factors, including the complexity of the physical processes and the many variables involved in the performance assessment (Tanasic and Hajdin, 2017), and the difficulties, costs and uncertainties associated with measurements of the consequences of bridge failure (Lamb et al., 2017). As a result of this, robust and validated methodologies for flood fragility and vulnerability assessment of bridges are scarce, although some attempts to develop such methodologies were recently made, with the aid of expert judgement or numerical modelling. Lamb et al. (2017) put forward a formal elicitation process to identify bridge vulnerability factors, summarizing the current knowledge of the problem of scour from various experts in the field. Not surprisingly, the foundation depth, type and the level of uncertainty in the estimation of these quantities emerged as the most important factors that should be considered when assessing bridge flood vulnerability and risk. However, the bridge type was ranked only 16th as vulnerability factor, which is quite interesting given the very different behaviour and capacity to withstand scour of a masonry arch bridge compared to a bridge with a multi-span simply supported deck. In general, modern steel and reinforced concrete structures, often founded on piles, should have been designed to withstand hydrodynamic forces and scour. They should also retain adequate vertical bearing capacity even under significant exposure of the total pile depth, provided the piles have a moment connection with the pile cap. On the other hand, masonry arch bridges are the most vulnerable to scour, due to the combination of their high stiffness and the fact that they are often built on shallow footings resting on the riverbed.

Among the numerical approaches investigating bridge vulnerability, worth mentioning are the works of Zampieri et al. (2017), Tubaldi et al. (2018), and Scozzese et al. (2019) on the simulation of the collapse mechanisms of masonry arch bridges with shallow foundations subjected to scour. Hydrodynamic forces are generally not a concern for these bridges unless the water level reaches the arch springing. In this case, there is also a potentially significant risk of debris accumulating at the bridge (Schmocker and Hager, 2011), resulting in further flow constriction, increased hydrodynamic forces, and higher scour rates and depths. If the water level exceeds the level of the arch soffit, the available hydraulic section is significantly constricted, not only in the horizontal but also in the vertical direction. Under these conditions, hydrodynamic forces become very significant and buoyancy forces may result in significant reduction of the vertical load carrying capacity (Hulet et al., 2006). Moreover, the vertical flow contraction exacerbates scour. More advanced and comprehensive numerical models and

methodologies need to be developed to assess the fragility of masonry arch bridges to the various flood actions. These models should account for the complex three-dimensional nature of the problem, as highlighted by studies investigating numerically the collapse mechanisms of some bridges (see e.g. Tubaldi et al., 2018; Wiggins et al., 2018; Scozzese et al., 2019).

Tanasic et al. (2013) developed scour vulnerability curves for a reinforced concrete bridge with a four-span continuous bridge, considering two failure modes, one related to the deformation capacity of the superstructure and the other to the bearing capacity of the soil-foundation system. Kim et al. (2017) also developed a methodology for flood fragility analysis of a multispan bridge considering multiple failure modes, including exceedance of pier or pile ductility capacity, pier rebar rupture, pile rupture, and deck loss. A recent study by Argyroudis and Mitoulis (2021) has focused on the vulnerability of prestressed concrete bridges to flood actions (scour, debris accumulation and hydrodynamic forces). Both integral bridges, where the abutment and piers are monolithically connected to the deck, and bridges with bearings were examined. Integral bridges were found to be more vulnerable to scour, since bearing flexibility in bridges with bearings provides some tolerance to scour-induced settlements. In addition, different structural components were found to be critical in different bridge types, e.g. the deck was found to be the most vulnerable structural component in integral bridges, and the bearings in the others, with settlements and hydrodynamic forces leading to serious damage of these devices. This shows that a substantial effort is needed to quantify the risk and the sequence of mechanisms that lead to the various bridge damage modes during floods.

Another strategy for vulnerability assessment is to infer fragility functions empirically from real-world (or perhaps experimental) loading and failure observations. In earthquake engineering, there are established statistical approaches for fragility assessment (Porter, 2015), the limit states and critical failure mechanisms of the resisting components are well defined, and there may be many observations of the limit state being exceeded within a single event; this is not the case for bridges and floods. Lamb et al. (2019) demonstrated the application of statistical inference to estimate a fragility function for railway bridge failures in Britain using observations of historical failure events, which were integrated within a whole-network economic risk analysis. The historical data could only be interpreted in this way by adopting a broad definition of bridge failure and expressing the intensity of the flood through its return period, a non-physical measure.

One major problem in bridge vulnerability assessment is the identification of a practical and representative intensity measure (IM) for quantifying the flood hazard and the vulnerability. For example, in Argyroudis and Mitoulis (2021) the maximum scour depth was used as IM, but scour can be a cumulative phenomenon, and thus there is only a mild correlation between scour depth and other actions (e.g. drag forces) during a flood. According to the outcomes of the study of Lamb et al. (2017), an appropriate intensity measure for the expression of bridge fragilities could be the flood return period itself. This could be helpful to summarize in a single scalar quantity the joint effects of different flood actions, but would not allow to fully decouple the hazard assessment from the bridge fragility evaluation. For example, one could consider the case of a bridge assumed to collapse when the water level reaches the deck. This bridge, placed in two rivers, one characterized by a significant flood hazard and the other by a low flood hazard, would present different probabilities of collapse (i.e., different vulnerabilities), since the probability of the water level reaching the deck would be higher in the case of significant flood hazard. This criticism of the use of the return period is ameliorated by the fact that modern bridges and scour protection

may respect design standards based on a specified flood return period, whilst older bridges are likely to embody some intuition about the local hazard, such that bridges will tend to be more resilient in locations that present a greater flood hazard. However, an underlying physical IM cannot be expected to scale linearly with the flood return period, so the increase in physical loading between, say, a 25-year and 100-year flood event could differ between locations, depending on their physical characteristics.

In addition, the return period typically only characterises the peak discharge, and not the duration of the event, which may be needed for the critical scour depth to develop. Thus, further studies are necessary in order to identify the optimal intensity measures for representing the flood hazard and quantifying the flood fragility for different bridge types. Alternatively, vector-valued IMs (Tubaldi et al., 2017b) could be considered for describing the flood hazard, e.g. combining the flow height/velocity, representing the hydrostatic and hydrodynamic force, and the maximum scour depth, representing the scour action. Fragility surfaces could be used for quantifying the probability of bridge failure conditional on multiple intensity measures, whereas state-dependent fragilities would be needed to account for the existing scour depth resulting from the action of past flood events. Further studies are also needed to identify the engineering demand parameters and limit states for the components of bridge types other than concrete ones.

Finally, another important aspect in the development of vulnerability curves for bridges is the characterization of the costs and consequences due to bridge performance degradation. These should include the direct consequences of structural damage (e.g. repair costs required to return the damaged bridge to its original state, as well as injuries, life losses, etc), and indirect consequences (e.g. service restrictions, additional travel time and travel distance costs for network road users). With regards to direct costs, worth to mention is the record prepared by Cumbria County Council of scour depths, bridge damages and repair costs resulting from the December 2015 floods in Cumbria for 350 sites. This record constitutes a unique opportunity to carry out a monetised assessment of the risks from extreme flooding. Combining this record with hind-cast flows for the storm events can provide a data set that can be mined to yield correlations between the cost of damage and all the site and flow variables. Some preliminary analyses performed by Mott MacDonald have shown that two thirds of the repair costs could have been avoided had it been feasible to identify and protect the most vulnerable 11% of the damaged bridges.

With regards to indirect consequences, while there are many tools available to assess the impact of bridge closure on the traffic flow in a network (see e.g. Liu et al. 2018; Lamb et al. 2019), there is a current lack of data on the high repair costs and downtimes associated with various bridge failure modes (Figure 3). Thus, recourse to expert elicitation appears unavoidable for characterising this, as discussed more in detail in the next subsection.

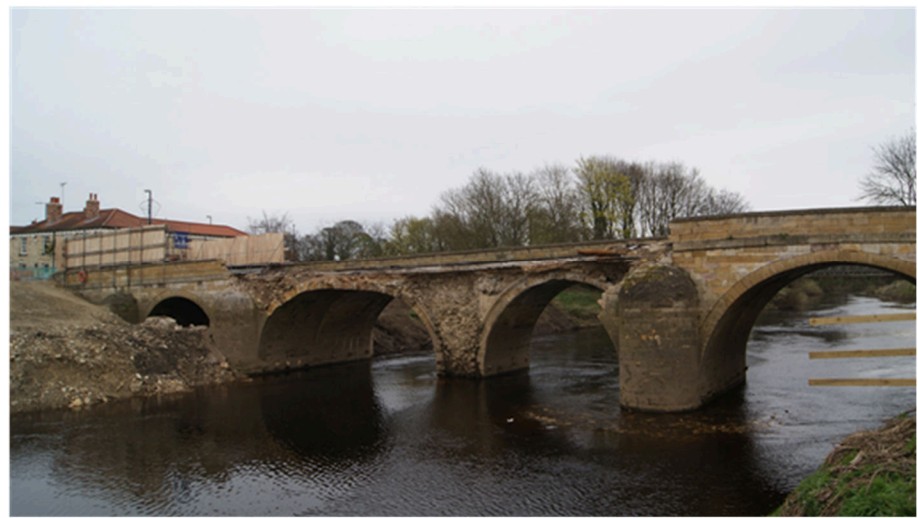

**Figure 3. Tadcaster bridge, damaged by Storm Eva in 2015, repaired and reopened to traffic 13 months after. Bridge closure resulted in 9-mile detour of 20 minutes to reach the other side. Source: https://commons.wikimedia.org/wiki/File:Tadcaster_Bridge_closed_following_last_years_damage_(10th_April_2016)_005.JPG**

## 2.4. Quantification of restoration and reinstatement models

The definition of the resilience of bridges to natural hazards such as floods and earthquakes is a matter of continuous debate, and there is no consensus on which tools and metrics to use or how and when to apply them. As pointed out in Alipour (2017), one of the key concerns regarding the definitions of resilience currently available is the over-emphasis on the pre-disaster side of the problem and the measures that aim to reduce potential capacity losses (i.e., rip-rap) (Badroddin and Chen, 2021), and the less attention given to the emergency response and recovery phases and measures following the disaster. However, in the authors' opinion, both aspects are significant, as both proactive and reactive measures need to be implemented to mimimise the impact of floods.

The ability to quickly restore bridges whose stability or functionality has been or might be impaired by floods is essential to improve the resilience of transport infrastructure. It is perhaps the most pressing challenge for road and railway operators who manage bridges. The challenge is related to the prioritisation of mitigation measures, due to limited resources prior and/or after extreme floods, and the uncertainties associated to future events, the bridge performance, and the emergency and post-emergency management.

Apart from the technical challenges, the communication of resilience to stakeholders, which can include for example resilience metrics based on the cost of traffic detour and $CO_2$ emissions (see *e.g.,* Smith et al., 2021) is the crux of bridge flood resilience. After solutions are delivered on paper, resilience communication should then enable stakeholders' understanding and therefore facilitate them to implement resilience practices in their everyday tasks and justify spending in an objective manner. There is an urgent need to communicate resilience among engineers, governmental bodies, local authorities and the general public. As noted in Minsker et al. (2015), resiliency requires public awareness and a clear communication about disasters and the operation of critical infrastructure during flood events.

The work of Mitoulis et al. (2021) summarises the main tasks for bridge recovery after floods. The paper is the summary of an elicitation survey the results of which were made available in Mitoulis and Argyroudis (2021). In this paper, bridge recovery is split into structural restoration and functionality reinstatement. Restoration includes

all structural measures to tackle structural damage, whilst reinstatement is related to non-structural loss caused by, e.g. debris accumulation and or water on the bridge deck. The study highlights several findings and a number of inadequacies and challenges for future research endeavours. The first finding is non-engineering, and related to the reluctance of operators to identify the urgent need for bridge and transport network recovery models. Moreover, short-termism and short-term responsibilities in bridge maintenance leave little space and funds for long-term investment, e.g. for adaptation to climate change and rapid socio-economic changes, dictating new investment.

The second outcome of the elicitation survey was that restoration tasks have (small or large) spatiotemporal dependencies, as well as logical dependencies and are similar to different bridges. The duration of each restoration task depends on the extent of damage. Hence, the same restoration task (e.g. FRP strengthening of the deck) would be more time-consuming when the damage is more extensive. There was a great discrepancy in the experts' opinions and follow-up meetings were required to obtain more information with regard to the duration of restoration tasks. It was also established that there is a strong correlation between restoration (capacity) and reinstatement (traffic/functionality) times. Generally, operators are striving to reinstate functionality as quickly as possible and open the bridge to traffic, rather than retrofit the bridge and restore its structural capacity. Reinstatement is important for the operator as the aim is to reduce the indirect costs due to bridge closures. Therefore, reinstatement time was found to be approximately half of the total restoration time, indicating that restoration proceeds while the bridge is open to traffic.

## 2.5. Current flood risk management procedures

In the UK, the CS 469 (Takano and Pooley 2021) (formerly BD 97/12 (Highway Agency, 2012)) and the EX2502 Procedure (HR Wallingford, 1993) are employed respectively by the highway authorities (National Highways, Transport Scotland) and railway authorities (Network Rail) and by their respective operating companies for assessing and managing the flood risk of existing bridges. Transport Scotland has also introduced a Scour Management Strategy and Flood Emergency Plan which documents their response to scour inspection, assessment, and flood mitigation measures. Alternative procedures for the risk assessment of bridge exposed to floods have also been proposed by other organisations in the UK. CIRIA has produced a comprehensive manual on scour at bridges (Kirby et al, 2015) which covers the scour risk management process, from the identification of bridge elements exposed to hydraulic action, to the prioritisation of scour-susceptible structures and selection of options. The manual has been recently updated with the Supplementary Guide, CIRIA SP171 (Kitchen et al, 2021) to include the latest knowledge from asset owners, industry practitioners and academics. Following the 2015 event, Mott MacDonald and Cumbria County Council jointly developed a warning system for damaged bridges, using the live feeds from Environment Agency river level gauges as a surrogate for river flow (Mathews and Hardman, 2017). This system uses records of damage in December 2015 and a percentage of the associated record level as a predictor of further damage. This system proved cost effective for the management of damaged structures, though it falls short of the requirements for a true risk-based system.

These procedures are mainly focused on the scour hazard, which poses the major risk to their assets. The estimate of the scour depth under a hypothetical 200 year return period is used to categorise the bridge assets and prioritize risk mitigation interventions. These estimates often result in excessive and unrealistic levels of the scour depth,

which should not be interpreted as expected levels of scour under a flood scenario. The 20% increase (uplift) applied to the design peak flow to account for climate change effects does not account for any particular time horizon or regional differences, which vary between +4% and +52% in current guidance for England (central estimates, 2080s, Environment Agency, 2021), thus resulting in further bias and uncertainty in risk estimation. Information on the bridge vulnerability and potential losses (e.g. cost of repairs, traffic disruption, and the financial consequences of death and injury) are disregarded or taken into account in a simplified way by means of some heuristic coefficients increasing the risk rating. Moreover, priorities based on one return period do not consider the cumulative risk arising from less intense, but more frequent events (Tubaldi et al., 2017a). This is an often overlooked, important aspect, considering that the estimated return periods of the floods that lead to failure of many bridges in the UK that failed in the last 150 years were below 100 years (Van Leeuwen and Lamb, 2014). Flood emergency management and decisions concerning costly bridge closures are based on the water level at the bridge exceeding some limits (e.g. flood level markers corresponding to a 200 year return period flood, see e.g. Transport Scotland's Scour Management Strategy and Flood Emergency Plan), which are often difficult to correlate to the actual risk of bridge failure. An alternative to methods that focus on a single return period is based on the integration of damages caused by events with different probabilities (expected annual damage), which may provide a more comprehensive picture of the risk profile, although it also disregards the cumulative effect of sequences of events (Tubaldi et al., 2017a).

Hence, it can be concluded that the decision-making by transport agencies is not based on the explicit evaluation of the flood risk of bridges and of the expected losses arising due to bridge failure, and disregards many of the uncertainties inherent to the hazard, the data and the models used for risk assessment (see e.g. Dikanski et al. 2018; Pizarro and Tubaldi 2019; Bento et al. 2020).

During the workshop, another important limitation of current risk management approaches emerged, i.e. the fact that they rely significantly on visual inspections (Moore et al., 2001; Jeong et al., 2018). These include underwater bridge inspections, and are carried out by divers at regular intervals to check the state of any bridge component but also during and/or following flood events (e.g. reactive structures safety inspections and special inspections). Visual inspections are characterized by many drawbacks. They can be affected by human error, subjectivity of the inspector, can be expensive and time-consuming. Moreover, underwater inspections cannot be carried during heavy flood events and can be conducted only after floods have receded, when scour holes may have been refilled, thus hiding the real hazard to which the structure has been exposed.

The issues outlined above, combined with the difficulties in obtaining information regarding the typology, geometry and state of bridge components (e.g. bridge foundation type, depth, etc.), may severely limit our ability to the identify bridges at higher risk of failure. Thus, current risk management approaches could be improved by adding a more explicit assessment of the actual bridge risk with due consideration of various sources of uncertainty affecting the problem and of the consequences of bridge damage. HR Wallingford (Roca and Whitehouse, 2012) has also developed a fully probabilistic approach for scour risk assessment that could be used to quantify the probability of bridge failure by accounting for the uncertain structural response through bridge-specific scour fragility curves. A similar approach was advocated by Tubaldi et al. (2017) and Pregnolato (2019). Sasidharan et al. (2021) also developed a conceptual risk-informed approach for bridge scour management that considers the direct and indirect consequences associated with closure or failure of bridges due to scour within the decision

making. This framework could be used to identify cost-effective solutions for bridge scour risk management and mitigation. It could also be extended to allow selecting the most appropriate scour protection measure among the many available (see e.g. Kirby et al., 2015).

## 2.6. Use of inspection, monitoring and forecast data

Bridges and riverbeds are periodically assessed within general and principal inspections, and via reactive inspections following flood events. These inspections may provide some information on the temporal evolution of scour and of the bridge state, but it is only by resorting to environmental or structural health monitoring (SHM) measurements that damage can be anticipated or assessed in real time. Monitoring data can significantly contribute to increasing the resilience of critical infrastructure under a wide range of hazards by providing information useful for disaster prevention, disaster mitigation, and disaster recovery (Honfi and Lange, 2015; Achillopoulou et al., 2020). In particular, monitoring data can be valuable for achieving a better understanding of the behaviour of critical infrastructure assets under extreme events, and for model calibration and updating at any level, from hydrological and hydraulic (Beven et al., 2005; Montanari et al., 2009; Briaud et al., 2014) to structural (e.g., Prendergast et al., 2018).

Moreover, SHM improves the knowledge of the current state of the asset, and provides bridge managers with useful information for prioritizing retrofit and risk reduction interventions. It can also be useful for bridge state assessment before, during or after extreme events (Maroni 2020). Obtaining information regarding the integrity of the structure in near real time has positive effects for the rapid response to these events and the recovery, starting from the rescue operations. Thus, it is evident that SHM data can be useful in overcoming some of the limitations of visual inspections, reducing their frequency and increasing their reliability with complementary information.

A wide range of sensors and sensing techniques has been developed in recent years to support bridge flood risk assessment (see e.g. Prendergast and Gavin, 2014; Maroni et al., 2020; Tubaldi et al., 2020; Achillopoulou et al., 2020, which are mainly focused on the scour problem). However, current practices for bridge flood risk management have not benefited from the advancements in the fields of flood and bridge monitoring, due to reasons such as the high capital and installation costs of sensors, the difficulty in post-processing the large datasets they produce, the challenges in interpreting sensor observations and in fusing data from different data sources (Wu et al., 2020), and a lack of a rigorous quantification of the benefits they bring in terms of better-informed decision making in bridge risk management. One way to overcome the cost limitation is to install monitoring systems only at critical locations, by extending the information gained at these locations to the other assets through the use of Bayesian Networks (BNs) (see e.g. Maroni et al., 2020). These probabilistic tools provide a graphical representation of the various variables involved in a problem (e.g. scour risk assessment for a set of bridges in a network), and of their conditional dependencies. BNs can be used to efficiently spreading inside the network the information from sensors, which is usually limited to few variables (i.e. nodes). Maroni et al. (2020) developed a BN-based framework for evaluating the scour risk for three bridges crossing the river Nith in Scotland, exploiting data from scour probes installed at a bridge (Figure 4a) and gauging stations. The framework has been subsequently extended to include observations from inclinometers or GPS receivers (Tubaldi et al., 2021), which may also be useful for assessing the bridge state. A further extension of the developed BNs is required to allow merging information with different temporal resolutions, such as bathymetry observations obtained during

inspections (every few years) and continuous measurements of flow height or surface velocity. Such an extension would also allow accounting for the results of inspections. Methodologies are also needed for using sensor data to support decision-making and for quantifying the benefit, in terms of better-informed decision making, of the information provided by sensors. Concepts such as the value-of-information and the reduction of relative entropy could be used for this purpose (Giordano et al., 2020; Tubaldi et al., 2021), whereas theories such as expected utility (Cappello et al., 2016) and multi-criteria decision making (Triantaphyllou, 2020) could help to set sensor reading thresholds and configure alert settings. The criteria could be defined operationally, by asset owners, or through wider analysis of the number of users who may be directly or indirectly disrupted by the failure of physically interdependent infrastructures (see Thacker et al., 2017).

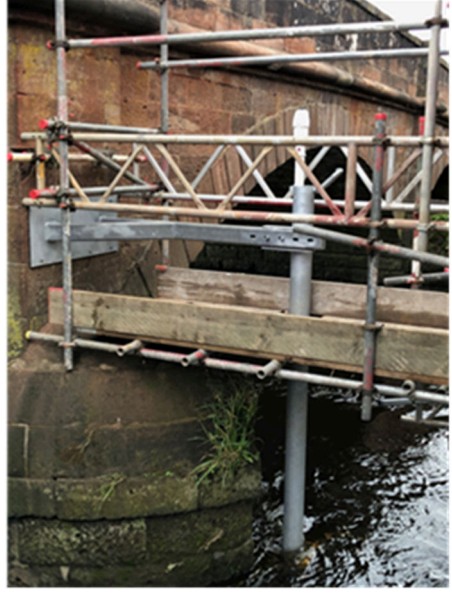
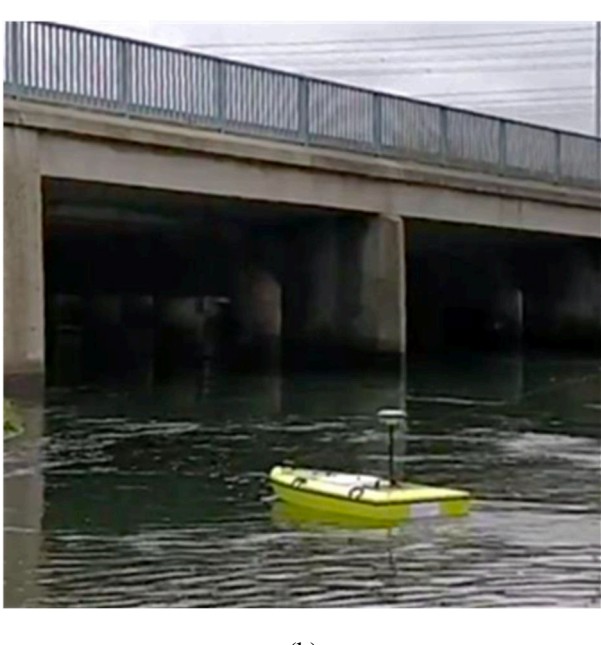

(a)                                                                 (b)

**Figure 4. a) Probe for continuous monitoring of total scour at a pier of the A76 200 bridge over the River Nith in New Cumnock (Maroni et al. 2020); b) Surveying remote controlled boat equipped with sonar, acoustic doppler velocity profilers, RTK-GPS and other motion sensors (developed at the University of Southampton).**

Accurate monitoring of scour depth during flood events is critical for emergency decision-making (closure and opening of bridges), but also to enable a more accurate, data-rich risk assessment strategy to be developed. Remote controlled survey boats (Figure 4b) may provide a relatively easy way to inspect critical assets during flood events. Based on the workshop and subsequent surveys, one field where more research work is required is the evaluation of the accuracy and the benefit of various techniques for the evaluation of the unknown foundation depth. This parameter, controlling the risk rating of bridges with shallow foundations, is often characterized by significant uncertainty. Although some non-destructive techniques have been proposed and are employed for the evaluation of bridge foundation depths (see e.g. Hossain et al., 2013; Tucker et al., 2015), they are not always accurate and reliable, and recourse to coring is often unavoidable. However, it is not infrequent that bridges classified at high risk of scour due to an initial conservative assumption of the foundation depth (e.g. between 0.3m and 1m for masonry bridges) are then considered at low risk level following a survey of the foundations. In many circumstances, it is advisable to spend more in accurate bathymetric surveys and extensive coring at multiple

locations if this permits to avoid installing expensive protection measures. This is for example the case when stones and material needed for riprap are not available on site, thus resulting in high transportation costs. On this regard, Network Rail has documented the case of a bridge where recourse to surveys of the riverbed has avoided deployment of scour protection measures.

Flood forecasting and monitoring sensors and data are vital for the future development of improved flood warning and risk monitoring systems. Impact-based forecasts - conveying information about the impact of the flood, taking into consideration vulnerability and exposure factors - for risk identification and communication have been shown to increase trust in warning systems, leading to more effective resilience building (Merz et al., 2020). Increasing the ability of the hydrological community to engage with the future development of impact-based forecasts (and to use machine learning and artificial intelligence tools to build and augment impact models (Wagenaar et al., 2020)) would help to further accelerate this process. For example, rainfall data combined with rainfall-runoff modelling for watersheds of critical bridges can be used to provide actionable alerts that could inform emergency management and trigger bridge closures (e.g. Cranston and Tavendale, 2012). Other aspects of flood forecasting, such as surface water flood forecasting (e.g. Speight et al., 2021), and forecasts on longer lead times can also be explored to assess their potential utility and application for bridge resilience. For example, the next generation of forecasts on larger spatial scales, such as the European-wide EFAS flood forecasting system (Wetterhall and Di Giuseppe, 2018), or the global subseasonal-to-seasonal meteorological predictions (White et al., 2017), can be employed to extend existing flood forecast and warning capabilities. These approaches pose some challenges due to the large uncertainties in the predicted rainfall at longer lead-times (or even over a few hours for surface water flooding after intense convective rain storms, Birch et al., 2021), significantly amplified by rainfall-runoff models (see e.g. Komma et al., 2007; Yu et al. 2016).

River level and velocity monitoring systems could be used for real-time risk monitoring. In this regard, it is worth noting that many wireless low-cost techniques have been recently developed that could be employed to gain useful information on the river hydraulic properties (e.g. Rivertrack sensors for measuring water level (Rivertrack 2021)) or cameras for particle image velocimetry (see e.g. Dal Sasso et al., 2021a). This is particularly relevant for ungauged river locations. Satellite imagery, aerial photography and UAVs technology (Figure 5) can also be very useful for monitoring morphological changes in rivers (Akay et al., 2021; Dal Sasso et al., 2021b) that may potentially lead to increased risk for bridges (see e.g. Lagasse et al. 2012; Koursari and Wallace, 2019). They can be important when there are accessibility issues (e.g. roads closed/destroyed) following flood damage. In general, it would be preferable to deploy sensors that do not need to be installed underwater, since obtaining permits from environmental agencies to work in watercourses can be problematic. Recent advancements in terms of image-velocimetry for fluvial monitoring could be adapted to give additional information on the hydraulic properties of the flow at bridges. This information could then be used to feed scour models for forecasting or assessing in real (or near-real) time the risk of bridge failure due to hydraulic actions. Some successful examples of the use of remote sensing techniques to recover flow velocities and river discharge are from Le Coz et al. (2010), Pizarro et al. (2020b), Eltner (2021), Bandini et al. (2021), and Fulton (2020).

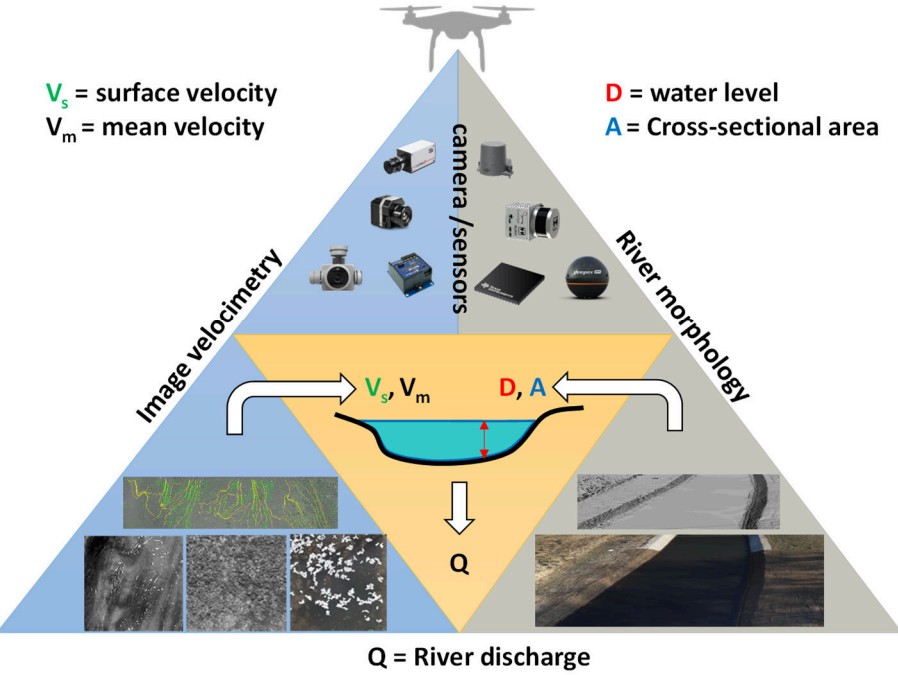


**Figure 5. Adapted from Dal Sasso et al. (2021b). Potential in the use of UAS for river monitoring combining river morphological and surface velocity estimations. The combinations of different sensors (e.g., RGB or TIR camera, LiDAR, eco-sounder, etc.) may help to measure flow at bridges in different fields and flow conditions.**

## 3. Conclusions and future directions

The workshop and subsequent meetings have highlighted significant gaps and uncertainties in bridge hazard assessment, vulnerability assessment and risk management. The gaps have a direct effect on the lifetime flood resilience of bridges, as illustrated in Figure 6. The uncertainty in the hazard leads to inaccurate models for the temporal occurrence of the flood events, and their intensity, with a direct effect on the expected levels of functionality drops. The uncertainty in the vulnerability results in the inability to predict the levels of functionality

drop under different hazard scenarios. Inaccurate procedures for identifying the bridges at risk due to flooding results in non-optimal allocation of resources for increasing robustness. Moreover, ineffective management procedures and lack of resources impede the speedy recovery and bounce back to bridge full functionality.

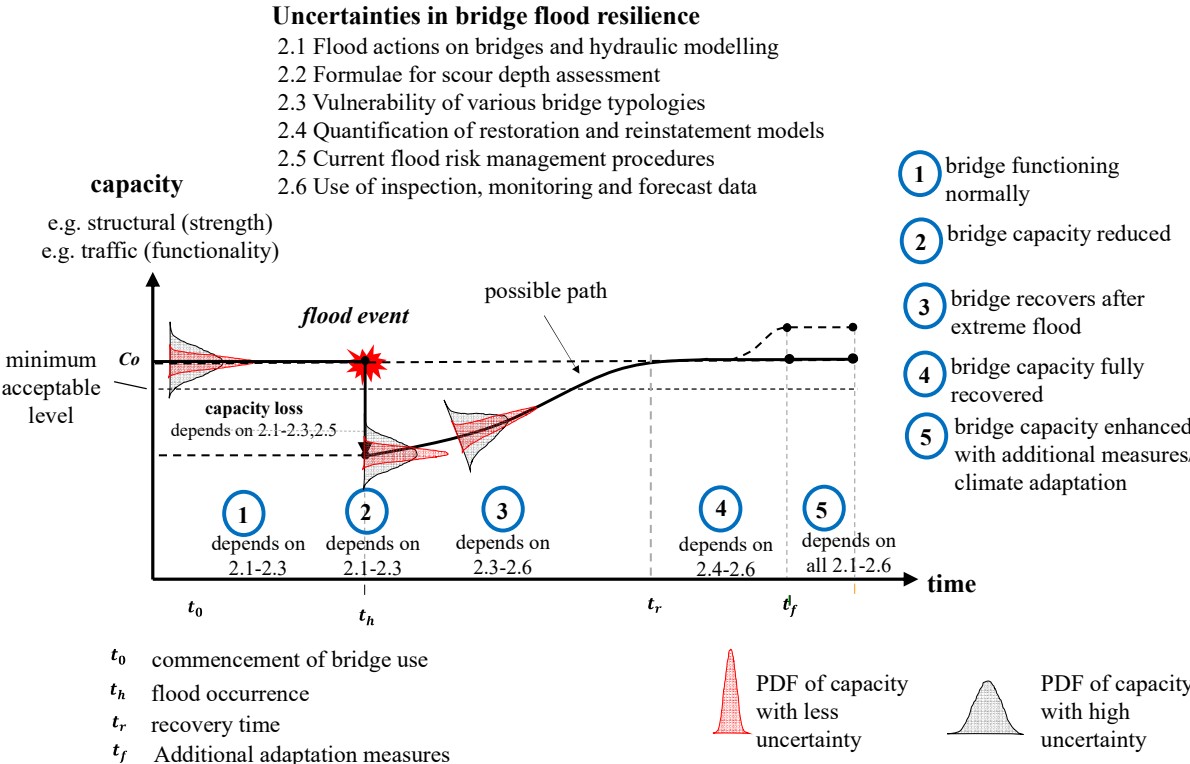

**Uncertainties in bridge flood resilience**
2.1 Flood actions on bridges and hydraulic modelling
2.2 Formulae for scour depth assessment
2.3 Vulnerability of various bridge typologies
2.4 Quantification of restoration and reinstatement models
2.5 Current flood risk management procedures
2.6 Use of inspection, monitoring and forecast data

**Figure 6. Contribution of uncertainties in hazard, vulnerability and emergency and post-emergency management to resilience.**


Table 1 provides a list of the most important research challenges, needs, and relevant actions that could contribute to the challenging goal of improving bridge resilience to floods. This list has been prepared taking into account the limited resources available to bridge owners and managers.

The actions outlined in Table 1 are expected to provide a manifold contribution to the various dimensions of life-cycle bridge resilience, namely robustness, resourcefulness, rapidity and redundancy (Mitoulis et al., 2021). More accurate models of the flood hazards and of the associated actions would help to understand and predict the causes of the drop in the performance of bridges in a context that could be significantly affected by climate change. Improved methodologies for evaluating the vulnerability of the components of different bridge typologies would

allow identifying critical elements and techniques for increasing their robustness. Better-informed rating systems and emergency and long-term risk-management strategies, accounting explicitly for the consequences of bridge failure and supported by forecasted and real-time monitoring data, can contribute to reduce the probability of bridge failure due to floods, the impact of the potential failure on transport networks and society, and the time to recovery. The results of the actions described in Table 1 can help shift the flood risk assessment paradigm from

manual and inaccurate diagnoses that rely heavily on costly and potentially inaccurate visual inspections, towards impact-based forecasting and near real-time evaluations of the risk supported by the fusion of data from multiple sensor technologies (Wu et al., 2020). This would ultimately help to accelerate the development of SHM-based digital twin platforms (Ye et al., 2019) for the management of bridges at risk of flooding, which are currently missing. It can also help to better define strategies to tackle the uncertain effects of climate change on the risk of

bridge failure due to floods.

In the near future, information from physical modelling and real-time data from heterogeneous sensors could be incorporated into the same platform to develop virtual representations of physical infrastructure assets that can be used to track the time-dependent state of the asset, with applications for both health diagnosis and prognosis (i.e. prediction of damage and functionality loss due to future events). Enforcing the Digital Twin concept in the context of flood risk assessment of bridges would provide infrastructure managers with valuable information, helping them to take optimal actions for both emergency response and long-term risk assessment and management. This could ultimately improve current risk management procedures which are overly simplistic and/or risky, and have not benefited from recent progresses in sensing and information technologies. The urgent need for this is widely acknowledged by the academics, bridge stakeholders and industry specialists that participated in the workshop on bridge flood risk assessment and management and contributed to this paper.

**Table 1. List of research areas, challenges & needs, and actions for improving bridge resilience to flooding.**

| Area | Research challenges & needs | Actions |
|---|---|---|
| **Hazard assessment and mitigation** | - Characterization of likelihood of debris accumulation at bridge piers.<br>- Critical evaluation of the effectiveness of technical solutions for mitigating hydrodynamic forces for bridges at risk of inundation.<br>- Extension of current flood forecast and warning capabilities to longer lead times and uncertainty characterization.<br>- More accurate modelling of the impact of climate change on frequency and intensity of flooding. | (1)-(5) |
| **Hydraulic actions modelling** | - Additional field research, data collection and analyses also needed to characterize the interrelated flood actions and validate models.<br>- Characterization of the temporal evolution of scour under the influence of time-varying intervening variables characteristic of flow and debris, with further experiments extending the range of applicability of developed approaches.<br>- Characterization of the effect of bridge pier and foundation geometries on the development of scour and on the scour hole shape.<br>-Development of models for establishing the relationships between measured river parameters (flow height, surface water velocity) and parameters controlling scour and hydraulic actions (e.g. depth averaged velocity). | (1)-(5) |
| **Vulnerability assessment and reduction** | - Identification of optimal intensity measures to be used in fragility analyses for describing the joint effect of various flood actions on bridges.<br>- Definition of methodologies for evaluating the vulnerability of various bridge types to concurrent flood-induced actions, accounting for cumulative effects (e.g. scour accumulated in previous floods) and the effects of debris through advanced modelling of water-soil-bridge assets.<br>- Statistics of the principal causes of failure and collapse mechanisms for various bridge typologies.<br>- Cost-benefit analysis of alternative solutions for mitigating the risk of different bridge components (e.g. deck unseating and uplift). | (1)-(6) |

| | | |
|---|---|---|
| **Risk management** | - Development of decision support tools to aid bridge managers to identify optimal actions for emergency/long-term flood risk management (including restoration and/or adaptation measures to climate change). These should take into account the bridge fragility and the consequences of bridge failure.<br><br>- Identification of actions that could be taken in the short term to mitigate the impact of forecasted floods (e.g. removal of debris accumulated at piers).<br><br>- More explicit considerations of structural vulnerability indicators and consequences in risk rating procedures.<br><br>- Improvement of response and recovery procedures that are kept up to date with the most recent technologies. | (2),(4)-(8) |
| **Impact-based forecasting** | - Tools enabling the paradigm shift from flood hydrograph to impact-based forecasting, so that mitigation measures can be better planned and justified using cost-benefit criteria. This could contribute to an increased awareness of the actual risk of bridges and a better acceptance of mitigation measures by affected communities. | (2)-(5) |
| **Monitoring and data fusion** | - Evaluation of the metrological effectiveness of sensors for monitoring the effects of floods on structures.<br><br>- Development of approaches for integrating information from numerical models and heterogeneous sensing systems, providing observations and measurements of different parameters involved in the risk assessment.<br><br>- Incorporation of monitoring technologies into risk management procedures. | (2)-(8) |
| **Value of information of data** | - Quantification of the benefits, in terms of cost savings to bridge operators and ultimately to communities, of data and information from sensors. This requires the development of a methodology for comparing the value of information from systems characterized by different measured quantities, accuracy, and spatiotemporal resolution. This effort could help to increase the adoption of sensors for monitoring bridges and rivers by bridge managers and operators.<br><br>- Cost-benefit analysis of risk mitigation measures (rip-rap) vis-a-vis bathymetric surveys and accurate foundation depth evaluations for identifying the most effective scour management strategies in case of unknown foundation depths. | (2)-(8) |
| **Resilience quantification** | - Restoration models for different types of bridges and different operators (masonry arch bridges vs. multi-span concrete bridges, road or railway bridges).<br><br>- Life-cycle resilience metrics for multiple flood scenarios including climate projections | (4),(5),(6),(7) |

Actions: (1) Laboratory and in-field experiments; (2) Development of models and techniques; (3) Numerical analyses; (4) Pilot case studies; (5) Data collection (through monitoring or desk studies); (6) Academic-industry workshops and engagement events; (7) Engagement with general public; (8) Training of experts, inspectors, recovery teams.


**Acknowledgements**

This work was supported by funding from the National Centre for Resilience under grant agreement NCRR2021-003 (project title "Evaluating the benefit of structural health monitoring for improving bridge resilience against scour") and by the Scottish Road Research Board (project title "Decision Support System based on "adaptive" Flood Level Markers").

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

**Code and data availability**

Literature used to inform this invited perspective is set out in the reference list.

**Author contributions**

All authors participated to the workshop or subsequent meetings underpinning this invited perspective. Enrico Tubaldi wrote the manuscript with contributions from all co-authors. Significant contribution was given by
Gustavo de Almeida on section 2.1, Alonso Pizarro on section 2.2, Rob Lamb on section 2.3, Stergios Mitoulis on section 2.4, Eftychia Koursari on section 2.5, Christopher White and Jim Brown on section 2.6, Richard Mathews on sections 2.2, 2.3 and 2.6. All the coauthors reviewed the paper and provided additional perspectives that enhanced the final version.

**Competing interests**

The authors declare that they have no conflict of interest.