# Peer review of "Invited perspectives: challenges and future directions in improving bridge flood resilience"

_Natural Hazards and Earth System Sciences, 2021_

## Author Comment (AC1)

**Response to Independent Reviewer Comments RC1**

**Comment:** Challenges and directions of future research in bridge flood resilience are discussed in this perspective article, based on the results of a workshop and a survey organised by the University of Strathclyde with experts from different fields. The article is timely, well written and deserves publication. I have a few comments and suggestions for improvement:

**Reply:** We thank the Independent Reviewer for their appreciation of the merits of the work and for the suggestions for improvement.

**Comment:** Please highlight the innovation of the paper against the state-of the art. If this is an agenda-setting paper, please comment on the timescales and mechanisms needed for solving these issues/problems. Why do the authors believe these challenges still exist? Perhaps a table explaining the causes and suggested solutions would be useful. This could include technical (engineering) but also other factors (accessibility, lack in methods, financial (resources), organisational, governance etc) to improve the significance of this paper.

**Reply:**

The use of a workshop to elicit the opinions of interested practitioners is not in itself innovative, having been demonstrated in Lamb et al (2017). However, in this case the intention was to focus primarily on the aspects of the response of bridges to scour and flood loadings that are often overlooked. Thus, the selection of attendees was weighted more towards engineers with experience in the management, assessment and modelling of bridges subjected to flood induced damage. In particular, several of the attendees had direct experience of preparing for and responding to the major floods in North West England in December 2015. This perspective, which included comparison of actual damage with the level of risk identified by UK standards (Hardman and Mathews 2017), was seen as important in identifying the issues requiring the most urgent investigation.

We have added the following sentence (highlighted in yellow) to explain what in our opinion are the causes of the identified issues and challenges:

"The fact that bridges continue to fail at a very high rate and the severe disruptions caused by bridge closures due to floods demonstrates the issues and uncertainties associated with current procedures and practices for assessing and mitigating the flood risk. These issues are due to a combination of

factors, among which the lack of knowledge of the problem, the gaps existing between the advanced techniques and methodologies developed by researchers and the more practical approaches adopted in risk management procedures, the lack of adequate human and technical resources, significant budget constraints, the tendency to acknowledge and address issues only when they manifest themselves in a catastrophic manner and to suppress rather than resolve problems. An analysis carried out by the RAC Foundation (2021) on bridges managed by local highways authorities in Great Britain has shown that there has been an apparent large decline in the number of bridges being assessed for risk of damage caused by river flow, despite 10 bridges fully collapsed and 30 partially collapsed in 2010. Thus, it is not surprising that the level of risk of many bridges exposed to flood effects remains largely unknown, with risk ratings still missing for many structures on secondary routes (more than 1000 structures in Cumbria County alone).

While efforts have been made to increase the robustness of bridges to withstand flood actions, transportation infrastructure managers face a unique challenge to prevent additional economic damage, often using maintenance budgets that are already stretched. For example, Transport Scotland spends £3-5m per annum on flood repairs and resilience works. The estimated cost to retrofit the 3,105 bridges managed by local councils classified as "substandard" is approximately £1 Billion (£985 million). However, budget restrictions mean that only 392 of these substandard bridges"

This is an agenda setting paper, intended to identify topics of research likely to yield the most significant benefits for bridge managers. The suggestion of detailing methodologies and timescales for resolving issues impacting bridge vulnerability is thus very helpful, although timescale can be very uncertain and strongly affected by governmental choices and allocation of resources for research and risk mitigation. Moreover, the research needs and challenges identified in Table 1 refer to various areas, where different experts and stakeholders should be involved (e.g. from hydrologists to structural engineers and bridge managers and inspectors). Thus, the various actions identified in Table 1 should and could be carried out in parallel. For these reasons, rather than adding an estimate of timeframes in the table, we have added a list of actions aimed at filling research gaps and addressing the identified needs and challenges. The revised Table 1 is attached at the end of this document.

**Comment:** It is suggested to discuss the challenges and knowledge gaps in the definition of sufficient risk and resilience metrics for flood critical bridges. Also, discuss the challenges in the communication of risk and resilience assessments to the stakeholders and decision makers.

**Reply:** Following the Reviewer's recommendation, we have added the following sentence:

The definition of the resilience of bridges to natural hazards such as floods and earthquakes is a matter of continuous debate, and there is no consensus on which tools and metrics to use or how and when to apply them. As pointed out in Alipour (2017), one of the key concerns regarding the definitions of resilience currently available is the over-emphasis on the pre-disaster side of the problem and the measures that aim to reduce potential capacity losses (i.e., rip-rap) (Badroddin and Chen 2021), and the less attention given to the emergency response and recovery phases and measures following the disaster. However, in the authors' opinion, both aspects are significant, as both proactive and reactive measures need to be implemented to mimimise the impact of floods. The ability to quickly restore bridges whose stability or functionality has been or might be impaired by floods is essential to improve the resilience of transport infrastructure. It is perhaps the most pressing challenge for road and railway operators who manage bridges. The challenge is related to the prioritisation of mitigation measures, due to limited resources prior and/or after extreme floods, and the uncertainties associated to future events, the bridge performance, and the emergency and post-emergency management.

Apart from the technical challenges, the communication of resilience to stakeholders, which can include for example resilience metrics based on the cost of traffic detour and $CO_2$ emissions (see *e.g.,* Smith et al. 2021) is the crux of bridge flood resilience. After solutions are delivered on paper, resilience communication should then enable stakeholders' understanding and therefore facilitate them to implement resilience practices in their everyday tasks and justify spending in an objective manner. There is an urgent need to communicate resilience among engineers, governmental bodies, local authorities and the general public. As noted in Minsker et al. (2015), resiliency requires public awareness and a clear communication about disasters and the operation of critical infrastructure during flood events.

The following references have been added:

Minsker, B., Baldwin, L., Crittenden, J., Kabbes, K., Karamouz, M., Lansey, K., ... & Williams, J. (2015). Progress and recommendations for advancing performance-based sustainable and resilient infrastructure design. *Journal of Water Resources Planning and Management*, *141*(12), A4015006.

Smith, A. W., Argyroudis, S. A., Winter, M. G., & Mitoulis, S. A. (2021). Economic impact of bridge functionality loss from a resilience perspective: Queensferry Crossing, UK. In *Proceedings of the Institution of Civil Engineers-Bridge Engineering* (pp. 1-11). Thomas Telford Ltd.

**Comment:** It is suggested to illustrate the impact of uncertainties on the hazard, vulnerability and restoration models, in the resilience assessment of flood critical assets. For example, show qualitatively how these uncertainties can change the resilience curve.

**Reply:** Following the Reviewer's suggestion, we have added the text below and Figure 6

[revised manuscript text omitted]

The revised Table 1 is attached at the end of this document.

**Comment:** Figure 5: please improve the figure, also explain the symbols Vs, Vm, D, A, Q

The figure has been improved, and an explanation of the symbols has been added.

[Figure]

**Comment:** Please check the citations, eg, line 244-245 should be Argyroudis and Mitoulis (2021) instead of Argyroudis et al (2021)

**Reply:** The citation has been corrected. Thanks for highlighting this.

[revised manuscript text omitted]

---

## Author Comment (AC2)

**Response to Comments CC2**

**Comment:**

The paper aims at exploring gaps around the robustness of bridges to the flood hazard on the basis of an expert workshop that took place in April 2021 with the participation of academics, consultants and decision makers operating in the United Kingdom. The topic is urgent and timely.

I understand that the paper sections are stated to be derived from a "workshop and subsequent meetings". However, the "workshop dimension" cannot be seen in the paper. For example, I was expecting to be told how the workshop was run and which information was sought (and how), how the participants were chosen and which expertise they were bringing to the table. The obvious reference here is Lamb et al. (2017) (https://nhess.copernicus.org/articles/17/1393/2017/nhess-17-1393-2017.pdf)/ . The paper reads more as an interesting review, since it does miss to link the outputs, techniques and discussion of the workshop to what is presented in the paper. It is not evidenced how Table 1 was obtained through the workshop and co-working, to give another example. Perhaps, it is worth to consider re-framing the paper within a "review" structure/perspective – or to report how to workshop was structured and how the paper's information was obtained.

Finally, the paper would have benefitted by discussing topics that were left out of the workshop (and could act as "future research"), such as netzero.

Minor comments below.

**Reply:** We thank the Reviewer for her comments and insights, which contributed improved the quality and clarity manuscript.

The workshop was run online during Covid restrictions. The academics from the University of Strathclyde, Surrey and Southampton that authored the manuscript invited their contacts and collaborators from the industry and transport agencies to attend the event. The main purpose of the event was for the academics to disseminate their latest research developments to the industrial partners, and to discuss about the existing knowledge and capability gaps in academia and industry with the aim of scoping a research agenda. Thus, the scope of the workshop was very different from that of Lamb et al. (2017), who focused on the uncertainties about the vulnerability of bridges to scour.

During the event, the partners introduced themselves and shared their experience, highlighting what were in their opinion their most challenging and urgent research needs.

It is noteworthy that not all the partners could attend the first workshop, and for this reason, some follow-up meetings were organised with some of the co-author of this study. The meetings and workshops were complemented by several exchanges of emails, where some partners shared additional thoughts and insights.

An online Word document was then created by Enrico Tubaldi, who also prepared a first draft of the manuscript with an initial structure. Each coauthor was given access to it and contributed to it. The manuscript underwent several revisions, until the final submission, which was approved by all the coauthors.

The section entitled "Author contributions" at the bottom of the manuscript highlights the contribution of the various co-authors to the document.

"*All authors participated to the workshop or subsequent meetings underpinning this invited perspective. Enrico Tubaldi wrote the manuscript with contributions from all co-authors. Significant contribution was given by Gustavo de Almeida on section 2.1, Alonso Pizarro on section 2.2, Rob Lamb on section 2.3, Stergios Mitoulis on section 2.4, Eftychia Koursari on section 2.5, Christopher White and Jim Brown on section 2.6, Richard Mathews on sections 2.2, 2.3 and 2.6. All the coauthors reviewed the paper and provided additional perspectives that enhanced the final version.*"

The authors believe providing additional information in the manuscript about how the workshop was run and further information exchanged would distract readers from the main goal of the manuscript, which is to highlight knowledge gaps, and cast the directions for future research and actions by academics, practitioners and bridge managers to improve bridge resilience to flooding. However, the response to this very interesting point raised by the Reviewer will be shared together with the document and could be accessed by the interested readers.

Regarding the choice of the format of the paper (i.e., Invited Perspectives article), our aim was not to provide a comprehensive state-of-the-art review of the problem, but to share "new ideas, views, or perceptions on a topical aspect of natural hazards", and "to stimulate an open debate among peers via the discussion phase", as per the guidance:

https://www.natural-hazards-and-earth-system-sciences.net/about/manuscript_types.html

Thus, following a consultation with the journal Editors, we decided that the manuscript would fall into the category of "Invited Perspectives" rather than "Review articles".

We also agree with the Reviewer that "netzero" is a very timely and important topic, and that it could have some links to bridge resilience. For example, extending bridges' design lifetime can contribute to netzero goals. However, this topic has not emerged during any workshop and subsequent meeting and thus we would prefer not to mention it in the paper.

**Comment:**

L47: "to be of the order of 160,000 in total with the Highways Agency …" may need rephrasing

**Reply**: The original sentence

".., but the number of bridges is estimated to be of the order of 160,000 in total with the Highways Agency (Middleton 2004), with about 30,000 of these crossing waterways"

has been rewritten as follows:

"The number of bridges managed by the Highways Agency is estimated to be as high as 160,000, with approximately 30,000 of these crossing waterways (Middleton 2004)."

**Comment:**

L47-50: what about all the other (non HA- or NR-owned) bridges?

**Reply:**

We don't have this figure, but we have provided additional information regarding road bridges managed by councils across Great Britain:

The estimated cost to retrofit the 3,105 bridges managed by local councils classified as "substandard" is approximately £1 Billion (£985 million). However, budget restrictions mean that only 392 of these substandard bridges will likely have the necessary work carried out on them within the next five years (RAC Foundation 2021).

**Comment:**

L64: used "£" for pounds before

**Reply:**

Corrected, thank you.

**Comment:**

L66: "this" what?

**Reply:**

We have rewritten the sentence as follows:

" The projected increase in winter precipitation and river flows due to climate change is expected to increase further the ==risk of bridge failure due to flooding== (Jaroszweski et al., 2021). ==This issue is also exacerbated== by.."

**Comment:**

L75: "the" Univ of Strathclyde (and also "the" for Surrey's and Southampton's)

**Reply:**

Thanks, we have added the "the" before "University".

**Comment:**

L133: used "and" instead of & before and after

**Reply:**

We apologise but did not understand this.

**Comment:**

L13: used "formulae" instead of formulas before

**Reply:**

Thank you for pointing this out, in the revised version of the manuscript we use only the term "formulae".

**Comment:**

L144, 243, 344, 431, 482, 493: "this" what?

**Reply:** We checked the use of this and believe there is no problem with it.

**Comment:**

L318: authors may want to refer to the updated version of BD97/12

**Reply**:

CS 469 has not been published yet. However, following the Reviewer's recommendation, we have added the following reference:

==Takano, H., Pooley, M. (2021). New UK guidance on hydraulic actions on highway structures and bridges. *Proceedings of the Institution of Civil Engineers – Bridge Engineering*, 174(3): 231–238, https://doi.org/10.1680/jbren.20.00024.==

**Comment:**

L460-3: how can "Satellite imagery, aerial photography and UAVs technology (Figure 5) can also be very useful…"?

**Reply:**

Morphological changes in rivers can led to aggradation, degradation, or lateral migration of the stream channel, all of which affect bridge scour. See e.g.

Brice, J. C. (1984). Assessment of channel stability at bridge sites. *Transportation Research Record*, (950).

Lagasse, P. F., Zevenbergen, L. W., Spitz, W., & Arneson, L. A. (2012). *Stream stability at highway structures* (No. FHWA-HIF-12-004). United States. Federal Highway Administration. Office of Bridge Technology.

The second reference has been added to the manuscript.

**Comment:**

L487-489: I think Digital twins are just an example of technology for this paper, rather than a conclusion of it (since there is no evidence before leading to it in this section).

**Reply:**

The sentence in the original manuscript:

"*This can help shift the flood risk assessment paradigm from manual and inaccurate diagnoses that rely heavily on costly and potentially inaccurate visual inspections, towards impact-based forecasting and near real-time evaluations of the risk supported by digital twinning technologies (Ye et al. 2019). It can also help to better define strategies to tackle the uncertain effects of climate change and socio-economic growth.*"

has been rewritten as follows:

The results of the actions described in Table 1 can help shift the flood risk assessment paradigm from manual and inaccurate diagnoses that rely heavily on costly and potentially inaccurate visual inspections, towards impact-based forecasting and near real-time evaluations of the risk supported by sensor technologies. This would ultimately help to accelerate the development of SHM-based digital twin platforms (Ye et al. 2019) for the management of bridges at risk of flooding, which are currently missing. It can also help to better define strategies to tackle the uncertain effects of climate change on the risk of bridge failure due to floods.

The revised Table 1 is attached at the end of this document.

**Comment:**

Table 1: what about the lack of flood damage models/curves for bridges at risk of flooding. What about resilience or restoration models? If these topics did not come out during the workshop, perhaps discuss anything that was left out during the event but worth to be mentioned? Also, maybe "data integration" rather than fusion?

**Reply:**

We agree with the Reviewers that there is a lack of flood damage models/curves for bridges at risk of flooding, as pointed out in Section "2.3. Vulnerability of various bridge typologies".

The actions we recommended in order to fill this knowledge gap are already in Table 1:

- Identification of optimal intensity measures to be used in fragility analyses for describing the joint effect of various flood actions on bridges.

- Definition of methodologies for evaluating the vulnerability of various bridge types to concurrent flood-induced actions, accounting for cumulative effects (e.g. scour accumulated in previous floods) and the effects of debris through advanced modelling of water-soil-bridge assets.

Regarding the problem of restoration models, this topic was not addressed in depth during the workshop. Stergios Mitoulis mentioned to the attendees his elicitation study, and it was established that recovery models for bridges affected by floods are generally missing. This leads to the absence of quantitative resilience models for various bridge types. The problem is discussed in the following references:

Mitoulis, S. A., Argyroudis, S. A., Loli, M., & Imam, B. (2021). Restoration models for quantifying flood resilience of bridges. *Engineering Structures*, *238*, 112180.

Mitoulis, S. A., & Argyroudis, S. A. (2021). Restoration models of flood resilient bridges: Survey data. *Data in brief, 36, 107088.*

The following item has been added in Table 1.

| | - Restoration models for different types of bridges and different operators (masonry arch bridges vs. multi-span concrete bridges, road or railway bridges). | (4),(5),(6),(7) |
|---|---|---|
| **Resilience quantification** | - Life-cycle resilience metrics for multiple flood scenarios including climate projections | |

The term and concept of "data fusion" is well established in the context of Structural Health Monitoring, see e.g.

https://en.wikipedia.org/wiki/Data_fusion

Thus, we would prefer to keep this term.

**Table 1. List of research challenges, needs and actions for improving bridge resilience to flooding.**

[revised manuscript text omitted]

- Life-cycle resilience assessments considering multiple flood scenarios under climate change effects. | (4),(5),(6),(7) |

Actions: (1) Laboratory and in-field experiments; (2) Development of models and techniques; (3) Numerical analyses; (4) Pilot case studies; (5) Data gathering (trough monitoring or desk studies); (6) Academic-industry workshops and engagement events; (7) Engagement with general public; (8) Training of experts, inspectors, recovery teams.

---

## Author Comment (AC3)

**Response to Independent Reviewer Comments RC2**

**Comment:** This paper examines the factor affecting bridge resilience and illustrates through literature research and the output of a recent workshop with major stakeholders in the sector possible actions to take based on the points individuated.

The paper is a welcome contribution given the importance of the infrastructure on the built environment and the future predicted impact of climate change, and present several important challenges and future opportunities. While the work is very comprehensive, there could be a few additional points of reflection that could be included:

**Reply:** We thank the Independent Reviewer for their appreciation of the merits of the work and for the suggestions for improvement.

**Comment:** In general, the text could be accompanied by more results from the literature, in particular of these could help highlight the elements of uncertainty.

**Reply:** Following the Reviewer's recommendation, we have added more references to the results in the literature, as discussed in detail in the responses to the following comments.

**Comment:** Line 135: One of the main factors of uncertainty in the scouring equation could be discussed further, examples include the definition of critical velocity (see Hamidifar et al 2021)

**Reply:** We agree with the Reviewer that empirical equations for scour assessment contains parameters such as the critical velocity whose definition is characterised by significant uncertainty. We have added the following sentences in the manuscript:

Artificial intelligence (in particular Machine Learning) is increasingly being used to produce more accurate multi-variate empirical predictors for scour (see e.g. Sharafi et al. 2016)…
..Another significant source of uncertainty affecting the estimation of the maximum scour depth is the evaluation of the flow critical velocity separating clear-water from live-bed conditions (Hamidifar et al. 2021).

The following references have been added:
Hamidifar, H., Zanganeh-Inaloo, F., & Carnacina, I. (2021). Hybrid scour depth prediction equations for reliable design of bridge piers. *Water*, *13*(15), 2019.

Sharafi, H., Ebtehaj, I., Bonakdari, H., & Zaji, A. H. (2016). Design of a support vector machine with different kernel functions to predict scour depth around bridge piers. *Natural Hazards*, *84*(3), 2145-2162.

**Comment:** Line 150: I would highlight also the first works by Oliveto and Hager 2002 and 2005, on temporal scour evolution

**Reply:** We agree with the Reviewer on the importance of the works of Oliveto and Hagers on the topic and have added them in the revised manuscript. The sentence:

"*Methods for time-dependent scour evaluations have been developed that can be applied for the assessment of scour under single (or multiple) flood events, opening the avenues for more accurate scour estimates. Additionally, and worthy of mentioning is the recent contributions for time-dependent scour modelling under non-stationary conditions. Among them, Pizarro et al. (2017a,b) and Link et al. (2017) proposed..*"

has been rewritten as follows:

"Methods for time-dependent scour evaluation have been developed that can be applied for the assessment of scour under single (or multiple) flood events, opening the avenues for more accurate scour estimates. The first studies on the topic considered the case of idealised hydrographs and clear-water conditions (see e.g. Oliveto and Hager 2002, Oliveto and Hager 2005), whereas more recent ones have also used more realistic hydrograph shapes. Recently, Pizarro et al. (2017a,b) and Link et al. (2017) proposed a model based on the dimensionless effective flow work, W*, for dealing with flood waves, and validated it against a wide range of unsteady conditions. Additionally, Link et al. (2020) proposed an extension of the model to consider the counter effects of erosion and deposition within the scour hole, which are typical of live-bed conditions."

The following references have been added:
Oliveto, G., Hager, W.H. (2002). Temporal evolution of clear-water pier and abutment scour. J. Hydraul. Eng. 128(9), 811–820.
Oliveto, G., Hager, W.H. (2005). Further results to time dependent local scour at bridge elements. J. Hydraul. Eng. 131(2), 97–105.

**Comment:** 175 -180 I would show some of the possible morphologies individuated from literature, which would help illustrate the point on important differences.

**Reply:** In general, morphologies are available for the simplified case of cylindrical piers. A recent work from Lee et al. (2020) has investigated more realistic pier and foundation shapes. The following text has been added in the manuscript:

"It is usually assumed that the shape of scour hole is indeed independent of the flow conditions and that it can be approximated by an inverted paraboloid with the upstream slope corresponding to the sediment's angle of repose, but these assumptions work well only for simple geometries such as cylindrical piers, as proven by Chreties et al. (2013), local scour conditions, and also for a flow direction perpendicular to the bridge longitudinal axis. Lee et al. (2021) have recently investigated experimentally the evolution of scour around piers and foundations with complex shape other than the cylindrical one, confirming that the maximum scour depth is attained upstream of the pier."

The following reference has been added:
Lee, S. O., Abid, I., & Hong, S. H. (2021). Effect of complex shape of pier foundation exposure on time development of scour. *Environmental Fluid Mechanics*, *21*(1), 103-127.

**Comment:** 210 Scouring on deck by Carnacina et al with debris accumulation also illustrate the potential increased scouring as well as flow acceleration (Carnacina et al. 2019 )

**Reply:** We agree with the Reviewer that this is a topic of extreme interest, as also emerged in a meeting between few authors of this study and Mark Pooley (Highways England). The following text and references have been added:

Another topic that is receiving considerable attention by researchers is the pressure-flow scour due to vertical contraction, which takes place in the case of submerged bridge deck (Carnacina et al. 2019). A recent review paper (Majid and Tripathi 2021) discusses the many research needs in this field.

The following references have been added:
Carnacina, I., Pagliara, S., & Leonardi, N. (2019). Bridge pier scour under pressure flow conditions. *River Research and Applications*, *35*(7), 844-854.

Majid, S. A., & Tripathi, S. (2021). Pressure-Flow Scour Due to Vertical Contraction: A Review. *Journal of Hydraulic Engineering, 147*(12), 03121002.

**Comment:** 220 Would be nice to have a comparison of various literature fragility curves, again as an illustration of the vast uncertainty existing around their determination also in line 265

**Reply:** Unfortunately, there are very few fragility curves available in the literature, and these cannot be compared because they refer to different bridge typologies and also they are based on different intensity measures. The problem of the definition of appropriate intensity measures is discussed in the paper in Section "2.3. Vulnerability of various bridge typologies" and research needs are highlighted in Table 1.

**Comment:** Line 410 the reference to Bayesian Networks is not fully clear? It seems an important challenge but the expected outcome could be extended further, how this could merge the data?

**Reply:** The text in the original manuscript:

"*One way to overcome the cost limitation is to install monitoring systems only at critical locations, by extending the information gained at these locations to the other assets through the use of Bayesian Networks. Criticality could be defined operationally, by asset owners, or take account of wider analysis of the number of users who may be directly or indirectly disrupted by the failure of physically interdependent infrastructures (see Thacker et al., 2017). This approach has been developed originally by Maroni et al. (2020) considering the problem of scour risk assessment, using data from scour probes (Figure 4a) and gauging stations. It has been subsequently extended to include observations from inclinometers or GPS receivers (Tubaldi et al., 2021), which may also be useful for assessing the bridge state. A further extension of the developed Bayesian Networks is required to allow merging information with different temporal resolutions, such as bathymetry observations obtained during inspections (every few years) and continuous measurements of flow height or surface velocity. Such an extension would also allow accounting for the results of inspections. Methodologies are also needed for using sensor data to support decision-making and for quantifying the benefit, in terms of better-informed decision making, of the information provided by sensors.*
*Concepts such as the value-of-information and the reduction of relative entropy could be used for this purpose (Giordano et al., 2020, Tubaldi et al., 2021), whereas theories such as expected utility*

*(Cappello et al., 2016) and multi-criteria decision making (Triantaphyllou, 2020) could help to set sensor reading thresholds and configure alert settings.*"

has been rewritten as follows:

"One way to overcome the cost limitation is to install monitoring systems only at critical locations, by extending the information gained at these locations to the other assets through the use of Bayesian Networks (BNs) (see e.g. Maroni et al. 2020). These probabilistic tools provide a graphical representation of the various variables involved in a problem (e.g. scour risk assessment for a set of bridges in a network), and of their conditional dependencies. BNs can be used to efficiently spreading inside the network the information from sensors, which is usually limited to few variables (i.e. nodes). Maroni et al. (2020) developed a BN-based framework for evaluating the scour risk for three bridges crossing the river Nith in Scotland, exploiting data from scour probes installed at a bridge (Figure 4a) and gauging stations. The framework has been subsequently extended to include observations from inclinometers or GPS receivers (Tubaldi et al., 2021), which may also be useful for assessing the bridge state. A further extension of the developed BNs is required to allow merging information with different temporal resolutions, such as bathymetry observations obtained during inspections (every few years) and continuous measurements of flow height or surface velocity. Such an extension would also allow accounting for the results of inspections. Methodologies are also needed for using sensor data to support decision-making and for quantifying the benefit, in terms of better-informed decision making, of the information provided by sensors. Concepts such as the value-of-information and the reduction of relative entropy could be used for this purpose (Giordano et al., 2020, Tubaldi et al., 2021), whereas theories such as expected utility (Cappello et al., 2016) and multi-criteria decision making (Triantaphyllou, 2020) could help to set sensor reading thresholds and configure alert settings. The criteria could be defined operationally, by asset owners, or through wider analysis of the number of users who may be directly or indirectly disrupted by the failure of physically interdependent infrastructures (see Thacker et al., 2017)."

**Comment:** Other general comments include:

Plots with the cause of failure of bridges as a percentage of mechanism, given the breadth of the stakeholders this statistic would be very welcomed

**Reply:** We agree with the Reviewer that this piece of information would be very important, but unfortunately, such statistics are not available. We have added this as a gap/need in Table 1 "List of actions and next steps for improving bridge resilience to flooding":

"- Statistics of the principal causes of failure and collapse mechanisms for various bridge typologies."

**Comment:** In the table and based on a high-level cost/opportunity analysis, which action should be taken first or prioritized, together with a desired temporal framework.

**Reply:** It is difficult to establish priorities and timeframes, since the research needs and challenges refer to various areas, where different experts and stakeholders should be involved (e.g. from hydrologists to structural engineers and bridge managers and inspectors). Thus, the various actions should and could be carried out in parallel. Moreover, timescales can be very uncertain and strongly affected by governmental choices and allocation of resources for research and risk mitigation. For these reasons, rather than adding an estimate of the timeframe in the table, we have added a list of actions aimed at filling research gaps and addressing the identified needs and challenges.
The revised Table 1 is attached at the end of this document.

**Comment:** Protections are generally overlocked, but several older bridges have been protected with rip-raps gabions, block ramps or similar structures.

**Reply:** We agree that protections have not been sufficiently described in the manuscript. However, we feel this is out of the scope of the Invited Perspectives article, which does not aim to provide an exhaustive review of available techniques. The CIRIA Manual provides a good overview of techniques and good references.
We have clarified this in the revised manuscript by rewriting the sentence:

"*This framework could be used to identify cost-effective solutions for bridge scour management and risk mitigation.*"

as follows:

"This framework could be used to identify cost-effective solutions for bridge scour ==risk== management and mitigation. ==It could also be extended to allow selecting the most appropriate scour protection measure among the many available (see e.g. Kirby et al. 2015).=="

**Comment:** I can't find Cantero-Chinchilla and de Almeida, 2021 in the list of references, which should illustrate the literature on debris impact on scouring. Other references on the topic exist that show the impact of debris accumulation on scouring, the impact on scouring protection that highlight important results on scouring temporal evolutions and morphologies (see for example the early work by Melville and Dongol 1992 but also Lagasse et al 2006, Pagliara and Carnacina 2010, Carnacina et al 2019)

**Reply:** We apologise with the Reviewer for the missing reference, which has been added in the revised manuscript:

[revised manuscript text omitted]